# Lectin Receptor-like Kinase Signaling during Engineered Ectomycorrhiza Colonization

**DOI:** 10.3390/cells12071082

**Published:** 2023-04-04

**Authors:** Him Shrestha, Tao Yao, Zhenzhen Qiao, Wellington Muchero, Robert L. Hettich, Jin-Gui Chen, Paul E. Abraham

**Affiliations:** 1Genome Science and Technology, University of Tennessee-Knoxville, Knoxville, TN 37996, USA; 2Biosciences Division, Oak Ridge National Laboratory, Oak Ridge, TN 37831, USA

**Keywords:** phosphoproteomics, lectin receptor-like kinase, plant–microbe interaction, symbiosis, ectomycorrhiza colonization

## Abstract

Mutualistic association can improve a plant’s health and productivity. G-type lectin receptor-like kinase (PtLecRLK1) is a susceptibility factor in *Populus trichocarpa* that permits root colonization by a beneficial fungus, *Laccaria bicolor*. Engineering PtLecRLK1 also permits *L. bicolor* root colonization in non-host plants similar to *Populus trichocarpa*. The intracellular signaling reprogramed by PtLecRLK1 upon recognition of *L. bicolor* to allow for the development and maintenance of symbiosis is yet to be determined. In this study, phosphoproteomics was utilized to identify phosphorylation-based relevant signaling pathways associated with *PtLecRLK1* recognition of *L. bicolor* in transgenic switchgrass roots. Our finding shows that *PtLecRLK1* in transgenic plants modifies the chitin-triggered plant defense and MAPK signaling along with a significant adjustment in phytohormone signaling, ROS balance, endocytosis, cytoskeleton movement, and proteasomal degradation in order to facilitate the establishment and maintenance of *L. bicolor* colonization. Moreover, protein–protein interaction data implicate a cGMP-dependent protein kinase as a potential substrate of *PtLecRLK1*.

## 1. Introduction

Plants coevolve simultaneously with diverse microbial communities [1,2,3,4] and establish molecular mechanisms to either permit or prevent the establishment of a particular microorganism [5,6]. Because microbial interactions can benefit plant sustainability and productivity, it is important to understand the genetic and environmental factors that determine interactions and their outcome on plants and their surrounding environments. Understanding the ecological and evolutionary principles governing these interactions provides an opportunity to engineer microbes and plants to achieve more sustainable and productive ecosystems [7] and mitigate risks associated with introducing microbes into non-native environments [8,9].

Quite remarkably, recent studies have shown that a single plant host gene can be genetically engineered to selectively prevent [10] or permit colonization for a particular fungus [11,12]. How these ‘susceptibility factors’ evolved to functionally override all other levels of plant immunity is poorly understood. In a recent study, we applied quantitative trait locus (QTL) mapping in poplar, which is an important biofeedstock for pulpwood, lumber, and bioenergy, and identified a susceptibility factor implicated in fungal root colonization [12]. It was determined that *Populus trichocarpa* encode a G-type lectin receptor-like kinase (PtLecRLK1) that permits root colonization for *Lacarria bicolor*, a beneficial ectomycorrhizal (ECM) fungus that provides poplar soil nutrients and water in exchange for carbon. Most intriguingly, genetically engineering PtLecRLK1 into non-host plants (Arabidopsis and switchgrass) fully permits *L. bicolor* plant root invasion and the establishment of intracellular hyphae (referred to as Hartig net), a prerequisite for symbiosis [11,12].

Upon fungal recognition, plasma membrane (PM)-localized receptor-like kinases (RLKs) trigger coordinated signaling pathways for an extensive new transcriptional program in the plant host, particularly in the root, for cellular remodeling and metabolic alterations to accommodate the growing interaction [13,14,15,16,17]. A multi-omics assessment of *PtLecRLK1* transgenic switchgrass roots identified dramatic changes in host transcription and translation, and the concurrent changes in the metabolite abundance that occurred with *L. bicolor* colonization [11]. Engineering PtLecRLK1 into a switchgrass plant changes its susceptibility to *L. bicolor* by reprogramming the expression of transcripts, proteins, and metabolites associated with intracellular transport, nutrient assimilation, carbohydrate metabolism, cell cycle and wall organization, and defense-related processes. Yet, despite this advancement, it remains to be determined how PtLecRLK1 recognition of *L. bicolor* alters intracellular signaling to reprogram the host for the development and maintenance of symbiosis. Therefore, the goals of this study were to identify phosphorylation-based signaling associated with *PtLecRLK1* transgenic switchgrass roots and to develop a conceptual model for relevant signaling pathways. To this end, phosphoproteomics data were generated for wild-type and *PtLecRLK1* transgenic switchgrass roots two months post-inoculation with *L. bicolor*.

## 2. Method

### 2.1. Plant Fungal Growth and Proteomics Sample Preparation

Switchgrass *PtLecRLK1* transgenic lines were generated as described previously [11]. Transgenic and wild-type switchgrass were co-cultured with *L. bicolor* liquid inoculum. Two-month post-inoculation, root tissues were collected for mass spectroscopy with at least three biological replicates. Each replicate was flash-frozen and ground under liquid nitrogen. Samples were processed for mass spectrometry measurement as described previously [11]. Briefly, the samples were dissolved in lysis buffer containing 4% SDS in 100 mM ammonium bicarbonate (ABC) buffer along with 1X Halt Protease Inhibitor Cocktail (Thermo Scientific; Waltham, MA, USA). The sample mixture was subjected to boiling, sonication, and centrifugation. The supernatant was collected and reduced with 10 mM dithiothreitol for 30 min and subsequently alkylated with 30 mM iodoacetamide in the dark for 15 min. The proteins were then isolated through a chloroform–methanol protein extraction protocol outlined previously [18] and reconstituted in 4% sodium deoxycholate solution. The protein concentration was quantified using a NanoDrop instrument (Thermo Scientific). The proteins were then digested using two consecutive aliquots of sequencing grade trypsin for three hours and then overnight at 37 °C at the ratio of 1:75 (trypsin to sample protein). Once digestion was complete, SDC was removed through precipitation with 1% formic acid and washed with ethyl acetate. The resulting peptides were lyophilized via SpeedVac (Thermo Scientific), desalted on Pierce peptide desalting spin columns (Thermo Scientific), and resuspended in 0.1% formic acid. A portion of the tryptic peptides (15 μg) was allocated for previously published proteomics measurement [11], while the remaining peptides were lyophilized and then resuspended in the manufacturer-recommended buffer for phosphopeptide enrichment. Phosphopeptide enrichment was carried out using phosphopeptide enrichment kits (Catalog number: A32992). Finally, the enriched phosphopeptides were lyophilized and then resuspended in 0.1% formic acid for phosphoproteomics measurement.

### 2.2. LC-MS/MS Analysis and Proteome Database Search

All samples were analyzed using an RSLCnano UHPLC system (Thermo Scientific) coupled with a Q Exactive Plus mass spectrometer (Thermo Scientific). The peptides were separated using a biphasic column (strong cation exchange and reversed phase) connected to nanospray emitter with a 75 μm inner diameter that was filled with 25 cm of 1.7 μm Kinetex C18 resin. For the phosphoproteome measurement, a single 1 μg injection of phospho-enriched peptides was analyzed with a 180 min gradient at a salt cut of 500 mM ammonium acetate. The Thermo Xcalibur software was used to acquire MS data in data-dependent acquisition (DDA) mode with MS2 acquisition set at top 10. All mass spectrometer data were processed in Proteome Discoverer 2.4 using MS Amanda [19] and Percolator [20]. MS data were searched against the *P. virgatum* and *L. bicolor* reference proteome database from DOE Joint Genome Institute (JGI), supplemented with transgenic and common laboratory contaminants sequences. The MS Amanda parameters for phosphopeptide identification were set as follows: MS1 tolerance = 10 ppm; MS2 tolerance = 0.02 Da; missed cleavages = 2; static modification = carbamidomethyl (C, +57.021 Da); dynamic modifications = oxidation (M, +15.995 Da) and phosphorylation (STY) (+79.966 Da). At both the peptide and PSM levels, the false discovery rate (FDR) was set to 1%.

### 2.3. Data Analysis

To perform differential abundance analysis on phosphorylated peptides, the peptide table was exported from Proteome Discoverer. Then, peptides with phosphorylation modification were extracted from the peptide table. These data were Log_2_ transformed, and LOESS normalized using InfernoRDN tool previously published [21]. Additionally, the data matrix was mean centered across all conditions. Only peptides present in at least two out of three replicates (in any experimental conditions) were deemed valid for further analysis. Missing data were imputed using Perseus software [22], with random numbers drawn from a normal distribution with parameters: width = 0.3 and downshift = 2.8. The resulting matrix was subjected to Welch’s *t*-test followed by Benjamini–Hochberg FDR correction to evaluate differential abundant proteins between the experimental groups. Finally, the differentially abundant phosphopeptides were mapped to their respective proteins to identify differentially abundant phosphoproteins.

### 2.4. Bimolecular Fluorescence Complementation (BiFC) Assay

BiFC assay was performed in *Populus* protoplasts as described by Zhang et al., 2020 [23]. In brief, the CDSs of PtLecRLK1 and its substrate candidate proteins were cloned into CFP^c^ (pUC119-CD3-1068) and VENUS^n^ (pUC119-CD3-1076) vectors through Gateway cloning, respectively. A total of 10 µg of CFP^c^-PtLecRLK1 plasmids and 10 µg of VENUS^n^-substrate plasmids were co-transfected in *Populus* protoplasts. After 18–20 h dark incubation, the reconstructed YFP signals were detected by a Zeiss LSM 710 (Jena, Germany) confocal microscope. ZEN software 2012 SP5 (Jena, Germany) was used for image processing.

## 3. Results and Discussion

Transgenic expression of *PtLecRLK1* can convert non-host plant species to a host of *L. bicolor*. These transgenic plants can develop a hyphal network between plant cells, improve a plant’s fitness in marginal growth conditions, and downregulate pathogenic defense [11]. These findings imply the potential of engineering the mycorrhizal symbiosis for improving plant health or productivity using *PtLecRLK1*. To uncover how *PtLecRLK1* regulates beneficial plant–fungal interaction, we performed phosphoproteomics analysis in transgenic (host) and wild-type (non-host) switchgrass. Because this plasma-membrane receptor is predicted to recognize fungal-cell-wall-derived ligands to suppress plant immunity for long-term colonization, we posit that the resulting signaling cascades are not transient but persistent. Therefore, we sought to characterize the resulting changes in phosphorylation signaling associated with established mycorrhization 2 months post-inoculation.

Across the experimental conditions, 284,588 peptide spectrum matches (PSMs) were identified, out of which 75% had phosphorylation evidence (Figure 1A). These PSMs were mapped to 5140 phosphopeptides in 4469 unique modification sites across 2760 phosphoproteins (Appendix A). A majority (87%) of these sites belong to amino acid serine, 12% belong to threonine, and the remaining portion belongs to tyrosine (Figure 1A). Most modification sites had a localization probability score of >90% (Figure 1A). A Welch’s *t*-test with an FDR correction at q < 0.05 and absolute log_2_ fold change greater than 1 was implemented to identify phosphopeptide abundances that differed between transgenic *PtLecRLK1* roots and wild-type (WT) roots during *L. bicolor* interaction. This quantitative analysis identified 1257 differentially abundant phosphopeptides (Figure 1B), of which 610 and 650 phosphopeptides were significantly up- and down-regulated in transgenic *PtLecRLK1* roots compared to wild-type (WT) (Figure 1B) (Appendix A). These phosphopeptides correspond to 603 and 647 differentially abundant phosphoproteins, respectively. The interpretation of quantitative phosphoproteomics can be challenging because differential phosphorylation events could be confused by simultaneous changes in protein abundance. Therefore, proteins previously determined to be differentially regulated in this pairwise comparison [11] were compared against the proteins with a significant change in phosphorylation. This comparison identified 73 phosphorylated proteins that were also observed to have regulated protein abundances, suggesting that the majority of these differentially phosphorylated proteins are regulated exclusively at the post-translational level (Figure 1C). These 73 proteins impacted by several levels of regulation were excluded from the additional analyses. The KEGG enrichment analysis identified MAPK signaling, endocytosis, and phosphatidylinositol signaling as enriched pathways at FDR 0.05 among the proteins that were uniquely regulated at the post-translational level.

In general, a large number of the phosphorylation modifications occurred on proteins and residues that have been previously implicated in plant defense and symbiosis (Appendix A). For instance, we observed a change in phosphorylation for CERK1, which is one of the most studied RLKs in fungal recognition [14,24,25] because it recognizes chitin found in most fungal cell walls [14,24,25]. In Arabidopsis, *AtCERK1* has been mostly studied for its role in defense-related chitin recognition [24] where chitin recognition results in *AtCERK1* phosphorylation at amino acids S266, S268, S270, S274, and T519 [24]. In our study, LC-MS/MS measurements identified a phosphorylation in the *AtCERK1* homolog (Pavir.6NG335100) and this modification was only observed in transgenic roots colonized by *L. bicolor* (Figure 2A). Sequence alignment analysis shows that the identified S19/T20 phosphorylation aligns well with site S274 from *AtCERK1* (Figure 2B). Chitin-triggered plant defense mediated by CERK1 leads to a MAPK signaling cascade and our analysis identified several phosphorylated proteins involved in the MAPK signaling cascade, which were only observed in transgenic *PtLecRLK1* roots during *L. bicolor* interaction (Figure 2A). In general, this observation suggests that chitin-triggered plant immunity through CERK1 is active. It is plausible that these molecular signatures are a result of having a higher amount of chitin exposed to plant root cells due to enhanced root colonization by transgenic *PtLecRLK1*. Alternatively, it is possible that CERK1 is playing an active role in mediating *L. bicolor* symbiosis within transgenic *PtLecRLK1* roots. Recently, *OsCERK1* was implicated in a symbiotic relationship [15,26] and was shown to be necessary for promoting the colonization of AM fungi during symbiosis [15,26]. Unlike Arabidopsis, rice and switchgrass CERK1 homologs lack LysM domains necessary for chitin recognition. Therefore, it is plausible that the observed phosphorylation alters a coreceptor specific to enabling symbiosis [15,26]. Because we have previously shown that transgenic *PtLecRLK1* Arabidopsis roots can be colonized by *L. bicolor*, the presence or absence of the CERK1 LysM is less likely to be a crucial aspect of this engineered symbiosis and further work is needed to determine the impact of the observed protein modification.

The substrate(s) of PtLecRLK1 are currently unknown. To identify putative downstream targets, protein–protein interaction (PPI) information was collected for PtLecRLK1 (Potri.T022200; v3.1) from the STRING database [27,28]. A cGMP-dependent protein kinase (PKG) (Potri.018G084900) was the only PPI reported (Figure 3A). The homolog of this protein in switchgrass (Pavir.1NG172300) was uniquely phosphorylated in transgenic *PtLecRLK1* roots during *L. bicolor* colonization. To further assess whether this PKG is a substrate protein of PtLecRLK1, a bimolecular fluorescence complementation (BiFC) assay was performed in poplar protoplasts and the assay suggests PtLecRLK1 and this PKG interact with each other (Figure 3B). In plants, the role of PKG remains poorly understood. Unlike PKGs expressed in animals, those encoded in plant genomes are structurally unique because they contain an additional type 2C protein phosphatase (PP2C) domain [29]. PP2C-containing proteins are frequently shown to play crucial roles in biotic and abiotic stress responses, plant immunity, and plant development [30]. Recently, the Arabidopsis homolog of this PKG protein was described as an interacting protein of the calcium-associated protein kinase 1 (CAP1) and associated to root ammonium-regulated root hair growth [31]. Interestingly, four ammonium transporters (i.e., two isoforms of AMT1-1 Pavir.1KG399605; Pavir.7KG243500 and two isoforms of AMT 2 Pavir.9KG091401; Pavir.9NG008902) were significantly decreased in phosphorylation abundance in transgenic *PtLecRLK1* switchgrass roots when compared to WT. These AMT proteins are dynamically regulated, existing in either an active or inactive transporter state, and their activity is controlled by the phosphorylation of a conserved threonine residue in the C-terminus [32] (Figure 2B). Phosphorylation of threonine negatively correlates with root ammonium uptake [32]. The decreased phosphorylated protein abundance of all AMT suggests that transgenic *PtLecRLK1* roots during *L. bicolor* colonization is increasing the uptake of ammonium. Inside the plant root cell, ammonium is assimilated into glutamine with the help of glutamine synthetase (GS; Pavir.9KG542200) (Figure 2B), and glutamine acts as a key nitrogen (N) donor for cellular N metabolism and storage. Phosphorylation of GS has been shown to substantially decrease GS activity [33]. Intriguingly, our analysis showed a significant decrease in the phosphorylation of GS in *L. bicolor*-inoculated transgenic plants compared to WT, suggesting higher GS activity in the transgenic plant compared to WT. Regulation of glutamine in transgenic *PtLecRLK1* roots is further corroborated by the previous metabolomics analysis that showed an increased glutamine abundance in transgenic *PtLecRLK1* switchgrass roots when colonized by *L. bicolor* [11]. As such, these results lend support to *L. bicolor* playing a role in host ammonium acquisition and nitrogen metabolism, which is to be anticipated for ECM symbiosis, and provides insights into concomitant cellular reprogramming post-invasion.

To further advance our phosphorylation network, PPI information was then collected from the STRING database for PKG. In contrast to our PtLecRLK1 search, there are a much larger number of probable substrates (71 interacting partners identified in poplar with STRING experimentally and co-expression determined score of >0.90), and this suggests that PKG may be a hub kinase for downstream signaling (Appendix A). Interestingly, among those predicted substrates is a recently discovered susceptibility gene expressed in wheat that has been exploited by a fungal pathogen resulting in stripe rust infection [10]. It has been shown that fungal invasion results in the phosphorylation of TaPsIPK1 protein, which then enters the nucleus and phosphorylates CBF1d to increase fungal susceptibility [10]. Remarkably, inactivating this susceptibility factor has been shown to confer robust rust resistance in a field trial without a negative impact on growth and yield [10]. The switchgrass homologs of TaPsIPK1 (Pavir.1KG067500) and TaCBF1 (Pavir.2NG380900) were found to be uniquely phosphorylated in *L. bicolor*-inoculated transgenic switchgrass (Figure 2A,B).

In addition to these notable changes in phosphorylation status, our global analysis identified many other differentially abundant phosphoproteins related to plant defense, phytohormone signaling such as brassinosteroid signaling and ethylene response, ROS balance, endocytosis, cytoskeleton movement, and proteasomal degradation (Figure 2). Although it is outside the scope of this brief research communication to elaborately describe the implications of these findings for plant–fungal symbiosis, future studies can be targeted to interrogate the functional relevance of these pathways in depth for plant–fungal symbiosis.

## 4. Conclusions

PTMs, such as phosphorylation, represent a unique layer of regulation utilized by plants to adjust the molecular pathways necessary to either permit or prevent the establishment of a particular microorganism. Overall, this phosphoproteomics study facilitated the identification of phosphorylation-based relevant signaling pathways associated with *PtLecRLK1* recognition of *L. bicolor*. This rich dataset along with our previously published multi-omics data have helped to provide a more detailed understanding of how *PtLecRLK1* reprograms molecular pathways to facilitate the establishment and maintenance of *L. bicolor* colonization. Moreover, we detected an interaction and a putative PtLecRLK substrate that represents an exciting candidate for further interrogation of this signaling cascade. More broadly, this dataset can be used as a valuable resource for future research that focuses on cross-species comparisons to see if the *PtLecRLK1*-adjusted molecular pathways are conserved across multiple plant species. In practice, a deeper understanding of plant–fungal signaling pathways will be necessary to selectively engineer beneficial symbiosis while, figuratively speaking, leaving the door closed for pathogens.

## Figures and Tables

**Figure 1 cells-12-01082-f001:**
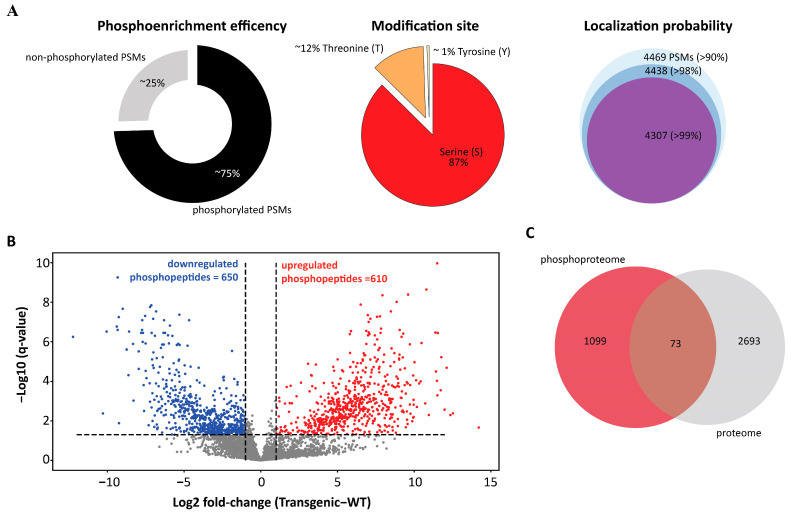
Qualitative and quantitative assessment of phosphoproteomics result from the root samples inoculated with *L. bicolor*. (**A**) Phosphopeptide enrichment efficiency, distribution of phosphosite, and localization probability of phosphorylation modification detected in our study. Most PSMs identified were phosphorylated. Most modified amino acid was serine and majority of the modified site has localization probability greater than 99%. (**B**) Volcano plot showing the differentially abundant phosphopeptides in our comparative phosphoproteomics analysis. Red nodes represent the phosphopeptides that were significantly increased in abundance in *L. bicolor*-inoculated transgenic line compared to WT and blue nodes that represent the phosphopeptides that were significantly decreased in abundance in *L. bicolor*-inoculated transgenic line compared to WT. Significantly changing phosphopeptides are those that pass q-value threshold of 0.05 and log2 fold change greater than absolute 1. (**C**) Venn diagram comparing the significantly changed phosphopeptides mapped proteins from phosphoproteome analysis to significantly changing proteins from proteome analysis.

**Figure 2 cells-12-01082-f002:**
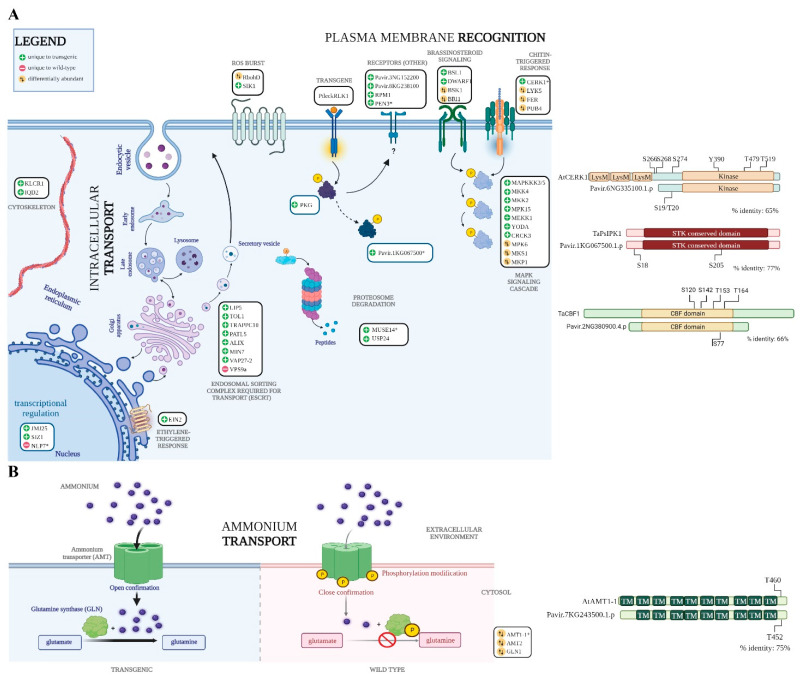
Signaling cascade and post-invasion molecular signature regulated as a result of PtLecRLK1 engineering in non-host plant. (**A**) Various signaling cascade mapped by significantly changed phosphopeptides in *L. bicolor*-inoculated transgenic line compared to WT. ‘+’ represent the phosphorylation unique to transgene, ‘−’ represent the phosphorylation unique to wild type, and ‘↑↓’ represent the phosphorylations that are identified in both transgene and WT but are differentially abundant. Some of the key proteins involved in plant–fungal interaction are shown with ‘*’ in accession and their simplified sequence alignments are shown on the right (see Appendix A for gene alias information). (**B**) Simplified model showing the regulation of ammonia uptake through ammonium transporter via phosphorylation modification. The phosphorylation of conserved threonine level negatively correlates with root ammonium uptake. Simplified sequence alignment for AMT1-1 is shown on the right with conserved threonine site.

**Figure 3 cells-12-01082-f003:**
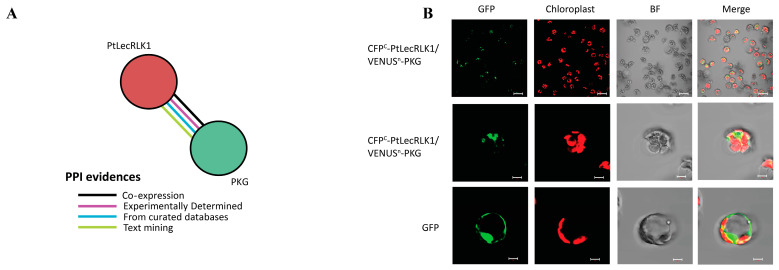
Identification and experimental validation of PtLecRLK1’s interacting partner. (**A**) Protein–protein interaction (PPI) obtained for PtLecRLK1 (Potri.T022200; v3.1) from the STRING database. Different lines of evidence for PPI are represented by different color edges. (**B**) Bimolecular fluorescence complementation (BiFC) assay performed in poplar protoplasts showed PtLecRLK1 and cGMP-PK (PKG) interact with each other as they produce green fluorescence. The scale bars for the top panel represent 40 µm, and the scale bars for the middle and bottom panels represents 5 µm.

## Data Availability

The mass spectrometry data from this study have been deposited at the ProteomeXchange Consortium through the MASSIVE repository at https://massive.ucsd.edu/; accessed on 22 February 2023. and can be located under the index MSV000091161.

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
