# Peer review of "Lectin Receptor-like Kinase Signaling during Engineered Ectomycorrhiza Colonization"

_cells, 2023, doi:10.3390/cells12071082_

Round 1

Reviewer 1 Report

In this study, the authors performed a phosphoproteomics analysis to investigate the signaling pathways associated with PtLecRLK1. A chitin-mediated signaling cascade was predicted downstream of PtLecRLK1. And the interaction between PtLecRLK1 and PKG was confirmed by BiFc in poplar protoplast.

However, I still have some questions about experimental design and data analysis.

1. The authors used root tissue at two months post-inoculation. Normally, the phosphorylation of protein is a transient event inside a cell. How to explain how phosphorylation is required for PtLecRLK1-mediated signaling?

2. The authors compared the proteome and phosphoproteome results to illustrate the true alternation of signaling inside plants. Overall, those differentially modified proteins are related to which pathways or biological processes? A KEGG or GO enrichment assay may give clues for it.

3. At the beginning of the result part, please briefly give an introduction to experimental design which helps readers to understand the biological process.

4. Does the WT and PtlecRLK1 mutant show significant differences with or without colonization of L. bicolor? This would help to understand the biological significance of PtlecRLK1 in the symbiosis process.

Author Response

In this study, the authors performed a phosphoproteomics analysis to investigate the signaling pathways associated with PtLecRLK1. A chitin-mediated signaling cascade was predicted downstream of PtLecRLK1. And the interaction between PtLecRLK1 and PKG was confirmed by BiFc in poplar protoplast.

However, I still have some questions about experimental design and data analysis.

  1. The authors used root tissue at two months post-inoculation. Normally, the phosphorylation of protein is a transient event inside a cell. How to explain how phosphorylation is required for PtLecRLK1-mediated signaling?

Yes, phosphorylation modifications are most often described as transient events and we expect that there are likely transient signaling cascades that occur upon initial fungal recognition were missed due to our experimental design. Yet, because our plasma-membrane receptor is predicted to recognize fungal-cell wall derived ligands and these are expected to persist over time during colonization, we sought to study the long term and stable changes in phosphorylation signaling to gain insight into establishment and maintenance of this engineered symbiotic relationship.

  1. The authors compared the proteome and phosphoproteome results to illustrate the true alternation of signaling inside plants. Overall, those differentially modified proteins are related to which pathways or biological processes? A KEGG or GO enrichment assay may give clues for it.

As recommended, we performed the KEGG enrichment. This has been added to the revised manuscript and can be seen in line 166-169.

  1. At the beginning of the result part, please briefly give an introduction to experimental design which helps readers to understand the biological process.

This has been added as suggested by reviewer in the revised version. This can be seen in line 138-149.

  1. Does the WT and PtlecRLK1 mutant show significant differences with or without colonization of L. bicolor? This would help to understand the biological significance of PtlecRLK1 in the symbiosis process.

There are system level changes as well as phenotype changes between WT and PtLecRLK1 expressed plant. These changes have been previously reported in https://doi.org/10.1111/pbi.13671.

Please find our revised manuscript attached.

Reviewer 2 Report

The manuscript titled "Lectin receptor-like kinase signaling during engineered ectomycorrhiza colonization" investigated the phosphorylation in the constitutive expressed PtLecRLK1 transgenic switchgrass roots under Laccaria bicolor co-culture condition. The analyses indicate that there have massive proteins involved in the protein phosphorylation during L. bicolor colonisation and symbiosis. The authors demonstrate the cGMP-dependent protein kinase (PKG) interacts with PtLecRLK1, and phosphoproteomics shows uniquely phosphorylated in transgenic PtLecRLK1 roots during L. bicolor colonisation. This will be an excellent source that will benefit the research for L. bicolor colonisation singling in the future.

 Some comments and suggestions:

 Line 226 - The authors state PtLecRLK1 and this PKG interact near the plasma membrane (Figure 3B). However, the green fluorescence signal in figure 3B seems to be located all over the cytoplasm. Not sure if the author has other images to show the signals close to the membrane? Theoretically, the PtLecRLK1 is located on the plasma membrane, and the protein interaction should also occur on the membrane. Please double-confirm the location or correct me if I am wrong.

 The author has confirmed that PtLecRLK1 and PKG interacted together. However, no experimental evidence to show their direct or indirect phosphorylation. Please provide in vitro phosphorylation assays to confirm that the PtLecRLK1 protein could direct phosphorylate the downstream target PKG.

 Suggest more experimental data needed for this manuscript to prove the finding through bioinformatics.

 Some text in Figure 2 is too small. Perhaps increasing the font size to make them legible and improve the figure quality (at least 300 dpi). The confocal image resolution for Figure 3B also needs to be increased (at least 300 dpi).

Author Response

The manuscript titled "Lectin receptor-like kinase signaling during engineered ectomycorrhiza colonization" investigated the phosphorylation in the constitutive expressed PtLecRLK1 transgenic switchgrass roots under Laccaria bicolor co-culture condition. The analyses indicate that there have massive proteins involved in the protein phosphorylation during L. bicolor colonisation and symbiosis. The authors demonstrate the cGMP-dependent protein kinase (PKG) interacts with PtLecRLK1, and phosphoproteomics shows uniquely phosphorylated in transgenic PtLecRLK1 roots during L. bicolor colonisation. This will be an excellent source that will benefit the research for L. bicolor colonisation singling in the future.

 Some comments and suggestions:

  1. Line 226 - The authors state PtLecRLK1 and this PKG interact near the plasma membrane (Figure 3B). However, the green fluorescence signal in figure 3B seems to be located all over the cytoplasm. Not sure if the author has other images to show the signals close to the membrane?Theoretically, the PtLecRLK1 is located on the plasma membrane, and the protein interaction should also occur on the membrane. Please double-confirm the location or correct me if I am wrong.

We thank reviewer for this feedback. We have removed “near the plasma membrane” from the revised version of the manuscript.

  1. The author has confirmed that PtLecRLK1 and PKG interacted together. However, no experimental evidence to show their direct or indirect phosphorylation. Please provide in vitro phosphorylation assays to confirm that the PtLecRLK1 protein could direct phosphorylate the downstream target PKG. Suggest more experimental data needed for this manuscript to prove the finding through bioinformatics.

Thank you for this comment. Based on the physical interaction we confirmed, we hypothesize that PtLecRLK1 directly phosphorylates PKG. However, confirming the direct regulation and validating its functional relevance will be the scope of a follow-up- research paper we are currently working on. We have qualified this finding in the main text and this revision can be found in line 358-360

  1. Some text in Figure 2 is too small. Perhaps increasing the font size to make them legible and improve the figure quality (at least 300 dpi). The confocal image resolution for Figure 3B also needs to be increased (at least 300 dpi).

We have increased resolution of the figures as suggested.

Please find the revised manuscript attached.

Round 2

Reviewer 2 Report

The authors have satisfactorily responded to all the comments and provided additional changes to the manuscript. The figure resolution has been increased. They all look good now.